# Comparison of Thermal and Laser-Reduced Graphene Oxide Production for Energy Storage Applications

**DOI:** 10.3390/nano13081391

**Published:** 2023-04-17

**Authors:** M. Belén Gómez-Mancebo, Rodolfo Fernández-Martínez, Andrea Ruiz-Perona, Verónica Rubio, Pablo Bastante, Fernando García-Pérez, Fernando Borlaf, Miguel Sánchez, Assia Hamada, Andrés Velasco, Yu Kyoung Ryu, Fernando Calle, Laura J. Bonales, Alberto J. Quejido, Javier Martínez, Isabel Rucandio

**Affiliations:** 1División de Química, Departamento de Tecnología (CIEMAT), Av. Complutense 40, 28040 Madrid, Spain; rodolfo.fernandez@ciemat.es (R.F.-M.); andrearzp@gmail.com (A.R.-P.); vrubiocarrera@gmail.com (V.R.); pablo.bastante@uam.es (P.B.); fernando.garcia@ciemat.es (F.G.-P.); fernando.borlaf@ciemat.es (F.B.); miguel.sanchez@ciemat.es (M.S.); alberto.quejido@ciemat.es (A.J.Q.); isabel.rucandio@ciemat.es (I.R.); 2Instituto de Sistemas Optoelectrónicos y Microtecnología, Universidad Politécnica de Madrid, Av. Complutense 30, 28040 Madrid, Spain; a.hamada@alumnos.upm.es (A.H.); andres.velasco@upm.es (A.V.); y.ryu@upm.es (Y.K.R.); fernando.calle@upm.es (F.C.); javier.martinez@upm.es (J.M.); 3Departamento de Ingeniería Electrónica, E.T.S.I de Telecomunicación, Universidad Politécnica de Madrid, Av. Complutense 30, 28040 Madrid, Spain; 4Unidad de Residuos de Alta Actividad, Departamento de Energía, CIEMAT, 28040 Madrid, Spain; laura.bonales@ciemat.es; 5Dpto. de Ciencia de Materiales, E.T.S.I de Caminos, Canales y Puertos, UPM, 28040 Madrid, Spain

**Keywords:** graphene-related materials, thermal and laser methods, reduced graphene oxide (rGO), energy storage

## Abstract

A way to obtain graphene-based materials on a large-scale level is by means of chemical methods for the oxidation of graphite to obtain graphene oxide (GO), in combination with thermal, laser, chemical and electrochemical reduction methods to produce reduced graphene oxide (rGO). Among these methods, thermal and laser-based reduction processes are attractive, due to their fast and low-cost characteristics. In this study, first a modified Hummer’s method was applied to obtain graphite oxide (GrO)/graphene oxide. Subsequently, an electrical furnace, a fusion instrument, a tubular reactor, a heating plate, and a microwave oven were used for the thermal reduction, and UV and CO_2_ lasers were used for the photothermal and/or photochemical reduction. The chemical and structural characterizations of the fabricated rGO samples were performed by Brunauer–Emmett–Teller (BET), X-ray diffraction (XRD), scanning electron microscope (SEM) and Raman spectroscopy measurements. The analysis and comparison of the results revealed that the strongest feature of the thermal reduction methods is the production of high specific surface area, fundamental for volumetric energy applications such as hydrogen storage, whereas in the case of the laser reduction methods, a highly localized reduction is achieved, ideal for microsupercapacitors in flexible electronics.

## 1. Introduction

Graphene has aroused great scientific interest, due to its unique properties: high thermal conductivity, excellent mechanical properties, enormous specific surface area of 2630 m^2^g^−1^ and exceptional electronic conductivity, among others. These properties make graphene attractive for a wide field of applications, including the fabrication of energy storage devices, which are the focus of this work [1,2,3,4,5,6]. Despite its huge potential, the commercial use of graphene is still limited, due to the high cost and low scalability of current production methods.

On the other hand, it is important to keep in mind that the fabrication process by which graphene is synthesized defines its properties, and as a result, its potential applications. Therefore, graphene fabrication is currently a highly researched topic [7], and the wide range of current synthesis methodologies include: exfoliation (either mechanical or chemical) [8,9], epitaxial growth through chemical vapor deposition (CVD) [10], unzipping of CNTs (via electrochemical, chemical, or physical methods), and the reduction of sugars (such as glucose or sucrose) [11], to name just a few. However, there is no single method of graphene synthesis that yields graphene exhibiting the optimum properties for all potential applications.

At present, the exfoliation of graphite constitutes one of the most promising approaches for obtaining large and scalable amounts of graphene-based materials with customizable properties, which depend on the synthesis conditions. The oxidative exfoliation process, in which graphite is oxidized under harsh conditions and subsequently exfoliated, can lead to the production of large amounts of graphene oxide (GO). Within these methodologies, wet chemical oxidation of GO can be performed using a wide variety of synthesis methods. The most popular method that is still used corresponds to the Hummers method [12], which involves the oxidation of graphite by mixing a slurry of graphite and sodium nitrite (NaNO_3_), concentrated H_2_SO_4_ and potassium permanganate (KMnO_4_). Several modifications of this method have been developed in the past 20 years, mainly to address some issues of the original Hummers method, such as replacing NaNO_3_ or achieving higher yield [13].

Production of reduced graphene oxide (rGO) from GO requires optimized procedures using strong reduction conditions (thermal, chemical, laser, electrochemical). For large-scale production, it would be desirable for the proposed method to combine ease of implementation, fast processing, low cost, and environmental friendliness. In this context, the thermal and laser reduction of GO are good strategies that yield more ordered graphene and additionally allow simultaneous reduction and exfoliation through the sudden expansion of CO or CO_2_ gases evolving to graphene sheets during the fast thermal heating of GO [14,15].

According to the literature, thermal and laser annealing of GO can effectively remove functional groups, depending on the applied temperature and the other experimental conditions such as time, power and pressure. Between 200 and 550 °C, hydroxyl and partial epoxy and carboxyl groups are removed [16]. Increasing the annealing temperature results in higher C/O ratios than those obtained using other reduction strategies [17]. This highly reduced graphene oxide allows the restoration of the conjugated structure and usually exhibits improved charge transport capabilities [18]. This makes the highly reduced graphene oxide suitable for high-quality electronic devices [19], sensors [20] and energy storage [21]. In addition, most reduction strategies require that GO must be dispersed in a solvent where sheets can freely move and they can be close to each other. This usually leads to agglomeration and layer stacking if spacers such as nano-fillers are not used [22]. In contrast, in thermal and laser reduction, GO sheets are dispersed in the solid phase, which reduces the extent of the sheet agglomeration [23]. In summary, thermal and laser reductions are easier and comparatively less expensive methods to obtain rGO, which makes them a feasible route for the bulk production of graphene.

In this study, different rGO materials were synthesized. First, a modified Hummer’s method was applied to obtain graphite oxide and graphene oxide (GrO/GO) from graphite powder (Gr). This method applied an oxidative treatment to graphite with potassium permanganate and sulfuric acid, achieving levels of oxidation of about 1:2 (C:O ratio). Then, the reduction from GO to graphene-like sheets by the removal of the oxygen-containing groups with the recovery of a conjugated structure was performed using several thermal and laser treatments. Five heating instruments: an electrical furnace, a fusion instrument, a tubular reactor, a heating plate, and a commercial microwave oven, were used for the thermal reduction. For the laser reduction, either an infrared (IR) CO_2_ laser or an ultraviolet (UV) diode laser were employed.

To evaluate chemical characterization, elemental analysis was performed by combustion. The X-ray diffraction (XRD) technique was employed for structural characterization. The as-prepared materials were assessed using Raman spectroscopy to evaluate the degree of rGO reduction and the presence of defects. The scanning electron microscopy (SEM) technique was also employed to characterize the morphology of the graphene nanostructures. Brunauer–Emmett–Teller (BET) and porosimetry tests were applied to determine the specific surface area (SSA) for all the as-prepared materials, as well as the pore volume and pore size distribution.

Previous works have validated the suitability of the thermal and laser reduction methods for the fabrication of graphene-based materials for energy storage applications, specifically hydrogen storage [24,25,26] and flexible microsupercapacitors [27,28,29,30]. In the present work, the correlation between the fabrication methods and the properties of the produced materials enables one to determine which method is more suitable for each of the aforementioned applications: the bulk synthesis nature of the thermal methods leads to the production of materials with higher SSA, ideal for volumetric applications such as hydrogen storage. The point-by-point, highly localized and superficial nature of the laser methods is ideal for in-plane and miniaturized applications such as flexible microsupercapacitors.

## 2. Materials and Methods

### 2.1. Materials

Concentrated sulfuric acid (H_2_SO_4_, 98% *w*/*w*), sodium nitrate (NaNO_3_), potassium permanganate (KMnO_4_) and hydrogen peroxide (H_2_O_2_, 30% *v*/*v*) were of analytical grade and from Merck (Darmstadt, Germany). Graphite powder (200 mesh, 99.9995%) was purchased from United Carbon Products (MI, USA). Ultrapure water (resistivity ≥ 18.2 MΩ·cm) from a Milli-Q system (Millipore Bedford, Bedford, MA, USA) was used throughout.

### 2.2. Synthesis of Graphite Oxide

Graphite oxide was prepared using a modified Hummer’s method from graphite powder. A total of 70 mL of concentrated H_2_SO_4_ was added to a mixture of 3 g of graphite powder and 1.5 g of NaNO_3_. The mixture was placed in an ice bath and stirred; 9 g of KMnO_4_ was slowly added, in small amounts, to keep the reaction temperature below 10 °C. After stirring for 1 h, the mixture was warmed to 35 ± 5 °C and stirred for 2 more hours. Then, 230 mL of ultrapure water was slowly added, while stirring and keeping the temperature below 70 °C. The resulting mixture was introduced into a water bath at 95 °C for 30 min, with manual stirring. Finally, 50 mL of H_2_O_2_ and, soon after, 200 mL of ultrapure water were added, resulting in a brilliant yellow color, along with bubbling and stirring for another 15 min. After that, it was left to decant overnight. 

Once the graphite oxide was decanted, the supernatant solution was removed and discarded. A total of 200 mL of 0.1M HCl was added to the solid and the mixture was stirred for 1 h. Subsequently, the mixture was filtered and the obtained powder was washed with 200 mL of ultrapure water, centrifuged at 10,500 rpm for 90 min, and the supernatant removed. This washing process was repeated until pH ≈ 5. Then the graphite oxide was collected, vacuum dried at 40 °C and ground in a tungsten carbide ball grinder.

Schemes that illustrate the oxidation process are described in Figure 1a.

### 2.3. Synthesis of Reduced Graphene Oxide

Two reduction processes were employed, using heat and light. The nomenclature given to the samples fabricated by each of the seven methods used in this work, which will be used throughout the text and the following tables and figures, are summarized in Table 1. 

Schemes that illustrate the synthesis of graphene oxide and the reduction processes using thermal and laser methods are depicted in Figure 1a, Figure 1b and Figure 1c, respectively.

A simple method for the production of thermally reduced graphene oxides (TrGO) is presented. Approximately 500 mg per batch of GO powder (prepared as indicated in Section 2.2) was taken for thermal exfoliation and reduction. 

Two different temperatures (250 °C and 450 °C) were tested and each sample was subjected to one of these temperatures for half an hour, on five different instruments: (a)An electrical furnace (EF-TrGO samples): the Tecno Piro 210 furnace was heated to the desired temperature and then GO powder was introduced and treated under air conditions.(b)A fusion instrument (FI-TrGO samples): GO powder was introduced into a platinum melting pot and all of it together was inserted into the fusion instrument model Perl’ X 3 (Malvern-PANalytical) at an automatic heating rate under air conditions.(c)A tubular reactor inside an electrical furnace (TR-TrGO samples): GO powder was inserted into the electric furnace of the reaction system Carbolite inside a quartz tube (20 mm outer and 17 mm inner diameters) under Ar atmosphere (100 mL min^−1^) until the desired temperature, at a rate of 50 °C min^−1^, and then held isothermally.(d)A heating plate (HP-TrGO samples): the IKA C-MAG HS 7 heating plate was heated until the desired temperature, and then the GO powder was placed on it and treated under air conditions.(e)A commercial microwave oven (Whirlpool) (MO-TrGO samples): In this case, a power control was applied instead of a temperature control. The GO powder was placed inside and two powers (320 W or 640 W) were applied, under air conditions.

In the case of the synthesis of rGO using laser reduction (LrGO), two different tools with different wavelengths were used:(a)A commercial continuous-wave CO_2_ laser (10.6 µm) with an XY control system was used to produce three samples of reduced graphene oxide (IR-LrGO samples). The laser power is up to 40 W and the scan speed can be tuned up to 600 mm·s^−1^. The sample is fixed, mounted on a static plate, and typically fixed with tape. Only the laser head moves at the specified speed; however, the used values rarely go over 100 mm/s. The focused beam size is around 100 µm. The maximum resolution of the machine is 25 µm. The graphene oxide powder in the water was sonicated for 2 h and then centrifuged at 3000 rpm for 20 min. The GO in aqueous solution (2.23 mg mL^−1^) was deposited as follows: a first drop casting of 100 mL was performed on commercial 16 × 25 cm^2^ acetate films with 100 µm of thickness (Crafter’s companion, Newton Aycliffe, UK). The films were then dried in air for 48 h. Next, a second drop casting of 100 mL was made on the same films to achieve a second layer of GO. Another 48 h was used to dry the films in air. Finally, the rGO samples were fabricated at a fixed scan speed of 100 mm·s^−1^ and a varying power of 1.6 W, 1.8 W and 2.0 W.(b)An ultraviolet diode laser (405 nm, 1000 mW), attached to an X-Y control system was used to produce three samples of reduced graphene oxide (UV-LrGO samples). The maximum resolution was 327 ppi (pixels per inch), and the only controllable parameter was the laser exposure time of each point, which could be varied from 1 up to 100 ms, in 1 ms intervals. As the machines that control each laser used in this work are different, the sample size that can be introduced into the machines are also different. In the case of the UV laser, the reduced dimensions of the machine require smaller samples. However, the amount of deposited material was calculated to produce the same thickness in proportion with the deposited area. The graphene oxide powder was sonicated for 1 h in water. Then, the GO aqueous solution (5 mg·mL^−1^) was deposited by means of three consecutive drop castings of 2 mL on glass sheets of 25 × 37 mm^2^, and dried in air overnight. Once fully dried, the samples were irradiated with the laser through two consecutive and complementary raster presets, to achieve complete areal irradiation. The first irradiation transformed a circular region in the center of each pixel, while the second irradiation directed the beam to the edges of the pixel. The rGO samples were fabricated using irradiation times of 2 ms, 10 ms, and 50 ms per pixel.

Figure 2a shows a picture of the GO powder before reduction (left) and a picture of the TrGO produced after HP treatment (right). In Figure 2b, a picture of the LrGO samples after the irradiation of the GO powder with the CO_2_ laser is shown.

### 2.4. Samples Characterization

X-ray diffraction (XRD) studies with Cu Kα radiation (λ = 1.54 Å) were performed with an X’Pert Pro diffractometer (Malvern-PANalytical) operating at 45 kV and 40 mA. XRD data were collected in θ-θ configuration in the angular range of 5 < 2θ < 80 with a 0.017 step size. 

The total carbon and hydrogen content was determined by combustion using a TruSpec CHN elemental analyzer (LECO, Benton Harbor, MI, USA). Carbon contents were determined by heating to a temperature of at least 900 °C in the presence of oxygen gas. Mineral and organic compounds were oxidized and/or volatilized to carbon dioxide (CO_2_). The amount of carbon dioxide was measured using an infrared detection method.

The specific surface area of graphene samples was determined using the Brunauer–Emmett–Teller (BET) method with an ASAP 2020 (Micromeritics, Norcross, GA, USA). In addition, the volume of pores of each sample was evaluated. 

Raman spectra were acquired using a Horiba LabRam HR evolution spectrometer (Jobin Yvon Technology, Edison, NJ, USA). The 532 nm laser beam was focused onto the sample through the 50× objective of an Olympus BX41 microscope. The scattered radiation was then collected in backscattering geometry, dispersed using a 600 grooves/mm holographic grating, and recorded using a CCD detector (256 × 1024 pixels), obtaining a resolution better than 0.48 cm^−1^/pixel. A typical spectrum was obtained within 60 s of acquisition time and five accumulations.

The morphology of the samples was observed using scanning electron microscopy, using a FEI InspectTM F50 (FEI Company, Columbia, MD, USA) at 5 kV of accelerating voltage.

## 3. Results and Discussion

XRD results are shown in Figure 3 and Table 2. A (002) diffraction peak appears in all cases between 23° and 27° (2θ), which is consistent with that of graphite 2θ = 26.43° (2θ) (PDF # 00-008-0415) [31]. This result corroborates the tendency of reduced materials to recover to the original graphite structure after graphite oxidation to GO, although the resulting fabricated materials show less crystallinity than graphite [32], which can be interpreted as a longer distance between the rGO sheets. From Figure 3 (Thermal samples) and Table 2, it can be seen that the (002) peaks of the samples prepared under thermal reduction methods lie between 23.3° and 24.5° (2θ). This narrow range of values shows that there are no significant differences between the different procedures used for the thermal treatment in this work. From the two different parameters used, 250 °C and 450 °C in the case of the EF-, FI-, TR- and HP- samples, and 320 W and 640 W in the case of the MO- samples, the angles are slightly higher for the higher temperature conditions, showing a more effective reduction degree. In samples MO-TrGO-320, UV-LrGO-2 and UV-LrGO-10, the presence of a more prominent peak at ≅10° (2Ɵ) corresponds to (001) GO reflection. This is due to a lower degree of reduction under these conditions [32]. 

In the case of the laser-assisted rGO samples, the UV-LrGO ones present the (002) peaks closest to the graphite structure (26.4°–26.6° (2θ)). The IR-LrGO samples also show a high level of reduction and a shorter distance between the rGO sheets, with (002) peaks in the range of 25.8°–25.9° (2Ɵ).

The distance between graphene layers was calculated by applying Bragg’s Equation (1) to (002) reflection:nλ = 2d sinƟ(1)
where n is an integer; λ is the wavelength of the incident X-rays; d is the lattice spacing (in this work, the rGO interlayer spacing), and Ɵ is the diffraction angle. An interlayer spacing of 0.336 nm corresponds to graphite [33]. After thermal reduction, the TrGO samples present an interlayer spacing in the range of 0.36–0.40 nm. In the case of the LrGO samples, the average d spacing corresponds to 0.34 nm. These results suggest that the oxygen and water existing between the layers could be removed through both thermal and laser processes, which, in addition to reducing GO to rGO, facilitate the exfoliation of bulk rGO into few-layer rGO, as happens in other materials to be used as 2D materials [34]. The reduction degree is more effective in the case of the laser treatments.

According to Scherrer’s Equation (2) [35]:D = κλ/βcosθ(2)
where D is the crystallite size (nm); κ is the shape factor; λ is the wavelength of the incident X-rays; θ is the Bragg diffraction angle; and β is the full-width at half maximum (FWHM). The FWHM calculated to (002) reflection (Table 2) could be used to determine the crystal size of the as-prepared materials. It can be seen that FWHM mean values in the TrGO samples are between 3.4° and 5.8° (2θ), while in the LrGO samples the FWHM values are between 0.27° and 1.7° (2θ). 

The average diameter of the stacking layer, denoted as D1 from Scherrer’s equation, with a constant equal to 0.9 [31], shows higher values in the LrGO samples, indicating an increase in the size due to the removal of impurities and crystallization (see Table 2). 

Another diffraction peak appears near 2θ = 43° in Table 2, corresponding to (100) reflection [36], and it indicates a short-range order in the stacked graphene layers. Scherrer’s equation was applied to this (100) reflection, with a constant of 1.84 [31], to evaluate again the average diameter of the stacking layers (parameter D2 from Scherrer’s equation), which present values between 13 and 71 nm in all samples. 

Taking into account all the parameters calculated by XRD, TrGO materials consist of a few graphene layers, in the range of 8 to 13, denoted n in Table 2, in a stacking nanostructure of an average diameter by height of approximately 4 nm × 14 nm, and a graphene layer distance of 0.38 nm. The number of layers obtained in the TrGO samples is similar to those obtained in other 2D materials [37]. For the LrGO samples, they consist of thick graphene layers, in the range of 28 to 190, in a stacking nanostructure of an average diameter of approximately 33 nm × 56 nm and a graphene layer distance of 0.34 nm. There is a significant difference between the two lasers, as shown in Table 2, with a smaller stacking nanostructure in the IR-LrGO samples. These IR-LrGO samples show similar numbers of layers to other vertical stacking heterostructures for optoelectronic devices [38].

Elemental carbon analyses of the synthesized samples are shown in Figure 4. These results can be interpreted as another reduction degree measurement. The TrGO materials show a carbon concentration in the range of 60–80%. This range of values shows that there are no significant differences among the different procedures used for the thermal treatment in this work. From the two different parameters used, 250 °C and 450 °C, in the case of the EF-TrGO, FI-TrGO and HP-TrGO samples, there are no significant differences among the different procedures used. However, in the cases of the TR-TrGO and MO-TrGO samples, higher temperature conditions resulted in higher carbon content, showing a more effective reduction degree. MO-TrGO-320, UV-LrGO-2 and UV-LrGO-10 show lower carbon content (Figure 4), since part of the GO has not been reduced to rGO, as shown in Figure 3.

The LrGO materials fabricated using UV laser present a carbon concentration in the range of 49–55%, presenting an increase in the concentration as a function of the exposure time. The LrGO materials fabricated using IR laser have a concentration in the range of 60–70%, presenting an increase in the C concentration as a function of power.

For all types of treatment, the elimination of functional groups occurs mainly due to CO and CO_2_ emissions. The process is enhanced both thermodynamically and kinetically in the presence of various oxygen functional groups located close to each other. The high oxygen density in graphite oxide/graphene oxide allows the removal of most of the oxygen present, even at low temperatures of 250 °C [32]. To eliminate the remaining oxygen functional groups, much higher temperatures (up to 1000 °C) or energy are required. However, we can point out that in those processes in which the thermal shock is higher, there is no temperature ramp or the temperature ramp is too steep, and the reduction processes are more efficient even at low temperatures, as can be seen in the EF-TrGO-, FI-TrGO- and HP-TrGO-samples. When the heat source needs to apply a temperature ramp to the sample, the reduction process is more effective in samples where a higher temperature is applied, as shown in the TR-TrGO-450 and MO-TrGO-640 samples. The highest C (79.7%) content was obtained in the TR-TrGO-450 sample, which shows that this process can be very effective at high temperatures. 

When a thermal shock occurs in these materials or when a temperature ramp is applied in which a higher temperature is reached (in this case there is also a greater temperature difference) the emission of CO and CO_2_ is very violent, which favors the expansion of the material and, as a consequence, increases the SSA. In cases where 250 °C is applied, with a temperature ramp, the applied temperature difference is lower, and therefore the emission of CO and CO_2_ is less steep and the rGO exfoliation is less effective.

LrGO materials have lower carbon concentrations than TrGO. Laser reduction methods provide a more intensive and focused source of energy. However, since this process is applied in separated spots and represents a surface technique [39], the amount of reduced material is smaller, compared to thermal methods.

Evaluation of the specific surface area in the TrGO and LrGO materials was carried out by the physical adsorption of nitrogen (N_2_) at 77 K and calculated according to the Brunauer–Emmett–Teller (BET) method. As it is shown in Table 3, the obtained SSA values are very different from one technique to other. This is caused by an explosive and uncontrolled process of exfoliation and reduction, in which gases (CO and CO_2_) are generated at a determined point, and this process is more or less expanded to other points; it is likely that the expansion of the material in this process can be highly dependent on the container. This method further shows the degree of exfoliation of the as-prepared rGO materials.

Due to the characteristics of each of the instruments used in thermal treatments, the only one that allows the use of the largest container and, therefore, allows a greater expansion of the material, is the heating plate. This fact could justify the larger specific surface area achieved with HP-TrGO-450 (680 m^2^.g^−1^) (Table 3). In general, most of the TrGO have SSA values (Table 3) in concordance with those previously reported [21,40,41], close to 500 m^2^.g^−1^, except for MO-TrGO-320.

Thermal treatments performed at 450 °C show higher SSA values than those performed at 250 °C. These results demonstrate that it is possible to obtain a highly porous rGO applying a very rapid heating system [40] and, as previously mentioned in the carbon content results, the thermal shock stimulates the exfoliation of the material, in this case GO, as demonstrated by the HP-TrGO-450 sample.

In the case of the LrGO samples, the SSA values obtained are smaller than those from the TrGO samples. As in the discussion of the C concentration, this result points out the superficial nature of the laser reduction process, where a smaller amount of reduced material is produced. When comparing the CO_2_ laser-irradiated samples, the IR-LrGO-2.0 one, fabricated with the highest power, presents less SSA than the IR-LrGO-1.8 sample. Since the laser beam represents a source of highly concentrated energy, the excess in the power parameter means the structural destruction of the material [42], hence the observed reduction in SSA. When comparing the UV laser-irradiated samples, given a fixed power value, the increase in the exposure time translates into a higher reduction degree and consequently, a higher SSA. Finally, the UV-LrGO samples for the longer exposure times show a higher SSA than the IR-LrGO samples. It is believed that this is related to the fact that for wavelengths in the IR and continuous wave mode, the dominant reduction mechanism of GO has been identified as photothermal, which is very efficient in the conversion of carbon into sp^2^ bonds, while in the case of wavelengths in the UV, depending on the used parameters, both photothermal and photochemical mechanisms can act [43,44]. The photochemical mechanism has been shown to be very effective in the removal of oxygen-containing groups. The presence of both mechanisms could enhance the reduction of GO more than the presence of only photothermal effect.

The porous texture characterization results corresponding to the materials prepared using thermal and laser reduction methods are also shown in Table 3. Microporosity analysis was performed on the samples using the t-plot model. 

The micropore volume results exhibit similar values for thermal and UV-laser-reduced materials, possibly due to the formation of crumpled and folder few-layered graphene sheets produced by applying a fast heating rate [45]. For the IR-laser-reduced materials and the thermal reductions with the smallest obtained SSA values, the smaller SSA obtained leads to the formation of a smaller network of micropores. Variation in porosity evolution after each treatment can influence the behavior of the materials in certain applications such as gas separation and water purification [46], DNA sequencing [47], hydrogen storage [48], supercapacitors [49] and heterogeneous catalysts [50].

The synthesized materials were further investigated using Raman spectroscopy. Three Raman spectra acquired at different points for each rGO sample were analyzed and compared. All spectra acquired for each sample were similar, thus confirming the homogeneity of the samples. From Figure 5, it can be observed that comparable spectra were obtained from all the samples produced by the different thermal (Figure 5, top) and laser (Figure 5, bottom) reduction treatments. All samples show two strong bands located at ~1350 cm^−1^ and ~1590 cm^−1^, assigned to the D and G bands, respectively [51,52]. These two bands are greatly broadened with FWHM around 100 cm^−1^, resulting in their overlap. The broadening in the D band may be due to the disorder/defects in the samples, but the broadening in the G band is unusual. 

Furthermore, at 2400 to 3250 cm^−1^ (represented as the 2D region), for all the TrGO (Figure 5, top) and LrGO (Figure 5, bottom) samples, the overtone 2D band peak around 2700 cm^−1^ and the second-order D+G band peak around 2940 cm^−1^ [53] were observed as bump-like features. 

The resolution level of the 2D and D+G peaks varies slightly, depending on the sample; however, the high level of broadness and convolution of the peaks indicates the multilayered structuring of the produced rGO materials and the presence of remnant functional groups and defects [54,55]. The remnant -OH, -H and -O functional groups are related to the degree of reduction, which is modulated by the processing conditions: temperature, time, atmosphere, wavelength and power. The incomplete removal of these functional groups leads to the existence of lattice disorder and, consequently, of Raman peaks associated with defects [56]. On the other hand, the sudden release of the CO and CO_2_ gases during the heating helps the exfoliation into thin graphene-like sheets. However, it also generates structural defects, due to the high amount of pressure exerted over the material in a short period of time [57]. Finally, in the case of the laser-reduced graphene oxide materials, the preparation method of the samples, namely the scratching from the substrate after reduction to obtain the powder, could have also contributed to the creation of structural defects. The reduction in the intensity of the Raman peaks related to defects, the increase in the intensity of the 2D peak and the narrowing of the FWHM of the D, G, 2D and D+G peaks can be achieved by optimizing further the reduction parameters of the seven methods employed in the present work.

The I_D_/I_G_ and I_2D_/I_G_ ratios for all samples are presented in Table 4. As is well known, the ratio of the D and G bands (I_D_/I_G_) is related to defect density [58]; thus, the I_D_/I_G_ ratio is larger when the surface has more defects. The I_2D_/I_G_ ratio gives information about the quality and number of layers of the graphenic material produced. From Table 4, it can be observed that the I_D_/I_G_ and I_2D_/I_G_ ratios of all samples are similar, contained within narrow ranges of 0.96–1.233 and 0.092–0.198, respectively. The values obtained for the I_D_/I_G_ ratios reflect the removal of functional groups upon reduction and the subsequent creation of defects [58]. The I_2D_/I_G_ ratios < 1 confirm that the materials consist of multilayer graphene [59,60]. The results obtained from Raman spectroscopy characterization show the multilayered nature and presence of remnant functional groups and defects in the generated rGO samples, reinforcing the conclusions obtained from the XRD analysis (Table 2).

It has been established that the presence of lattice defects in graphene sheets, which are a consequence of the thermal exfoliation route, causes the reduction in the elastic modulus and promotes the chemical reactivity and adsorption ability [61,62]. Vacancy defects also enhance the metal binding on graphene, increase the hydrogen uptake capacity, and change the binding energy into the desirable range for reversible hydrogen molecules adsorption [63]. High-defect-density structures have been shown to yield gravimetric hydrogen capacities as high as 5.81 wt% [63]. This rate value is close to the target specified by the US DOE (6.5 wt%) for hydrogen storage on board vehicles [64].

SEM microscopy was employed for morphological characterization. Most of the SEM images exhibit a well-expanded and exfoliated material with a porous structure (Figure 6). A quick release of gases generated during reduction treatments was observed for all treatments applied to the samples, causing not only reduction but also expansion and exfoliation, which could be a reason for their high SSA. This phenomenon was previously observed by other authors [14]. In this work, in both types of methods used to produce the samples, the expansion produced in the fabricated materials causes a similar morphology. A light material made up of separate and visible sheets corresponding to graphene is observed in both the TrGO and LrGO samples.

It can be noticed that there is a direct correlation between the SSA value and its morphology. Samples with higher SSA show images with larger, more separated, and translucent graphene layers. 

## 4. Conclusions

In the present study, two different types of treatments were performed, using a wide variety of instruments, to obtain reduced graphene oxides (rGO): thermal and laser methods. Both techniques enjoy the important advantages of being low-cost and fast. 

Thermal treatment was performed by means of five instruments with different characteristics: (a) an electrical furnace, (b) a fusion instrument, applying a rapid heating rate, (c) a tubular reactor, applying a slower heating rate, (d) a hot plate, applying a thermal shock, and e) a commercial microwave oven, applying radiation. All thermal treatments presented a similar reduction degree, with an average carbon percentage of 75%, a maximum of 13 graphene layers, an average diameter by height of about 4 nm × 14 nm, and a graphene layer distance of 0.38 nm, corroborated by a similar pore distribution with an average micropore volume of 0.0216 cm^3^. g^−1^. The specific surface area of these TrGO samples was mainly in the range of 244 to 682 m^2^.g^−1^. The main advantage of the thermal methods compared to the laser reduction methods is the larger values of pore volume, with higher specific surface areas, which makes it suitable for hydrogen storage applications, gas sensors, or porous support for heterogeneous catalyst systems.

Laser treatment was performed using a UV (405 nm) and a CO_2_ (10.6 µm) laser. The materials fabricated by both lasers present a lower reduction degree than the TrGO, with an average carbon percentage of 60% and with 28–190 graphene layers in a stacking nanostructure of an average diameter of about 33 nm x 56 nm and a graphene layer distance of 0.34 nm, with a larger particle size than in the TrGO samples, consistent with the smaller values of pore volume (average 0.0096 cm^3^. g^−1^) and smaller surface area, in the range of 49 to 133 m^2^.g^−1^. The main advantage of the laser reduction methods compared to the thermal ones is the ability to create patterns with free geometry, high precision, and high resolution on thin films. The simultaneous low penetration and highly localized characteristics of the technique make it ideal for flexible and miniaturized devices in micro- and nanoelectronic applications, such as flexible microsupercapacitors.

From this work, it can be deduced that it is possible to synthesize graphene materials with specific characteristics according to the application for which they will be intended. The analysis of the surface area, carbon content and the X-ray and Raman spectra identify the resulting material as if it were a fingerprint, which allows its best possible application.

## Figures and Tables

**Figure 1 nanomaterials-13-01391-f001:**
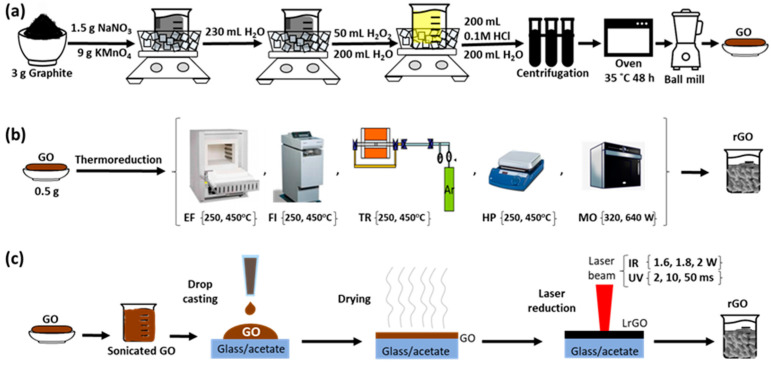
Schemes representing the process flow of the rGO preparation using (**a**) oxidation process, (**b**) thermal reduction and (**c**) laser reduction.

**Figure 2 nanomaterials-13-01391-f002:**
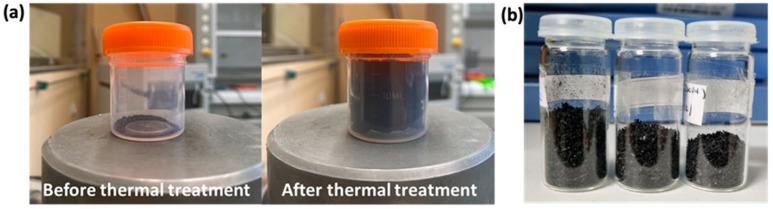
(**a**) Picture of the GO powder before reduction (**left**) and after reduction (**right**) using HP treatment. (**b**) Picture of LrGO material after reduction of GO powder using CO_2_ laser irradiation. Each bottle, from left to right, contains the rGO obtained by applying a power of 1.6 W, 1.8 W and 2.0 W.

**Figure 3 nanomaterials-13-01391-f003:**
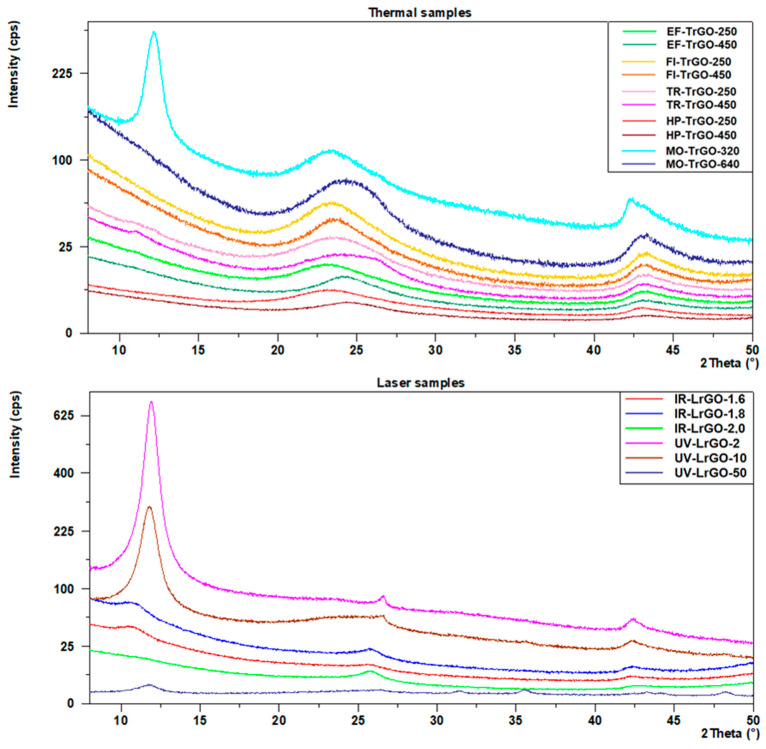
XRD peaks for the samples after the different thermal reduction treatments (**top**) and the samples after the different laser reduction treatments (**bottom**).

**Figure 4 nanomaterials-13-01391-f004:**
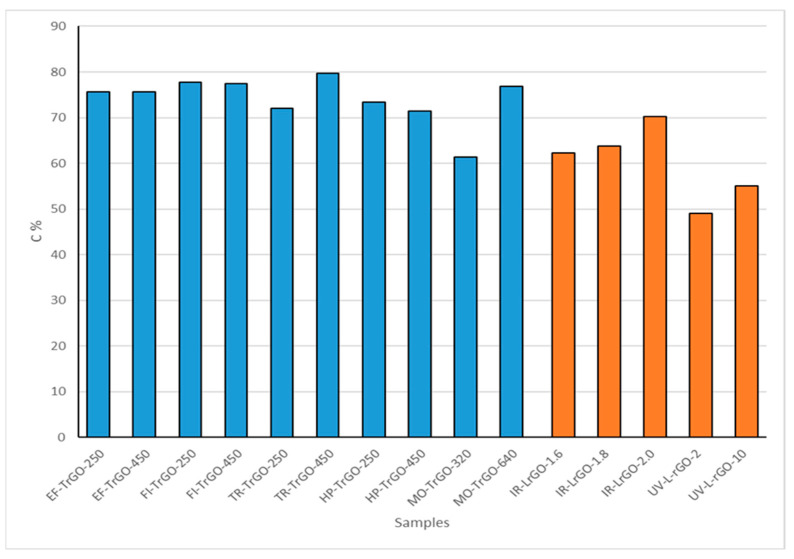
Carbon concentrations for TrGO and LrGO samples.

**Figure 5 nanomaterials-13-01391-f005:**
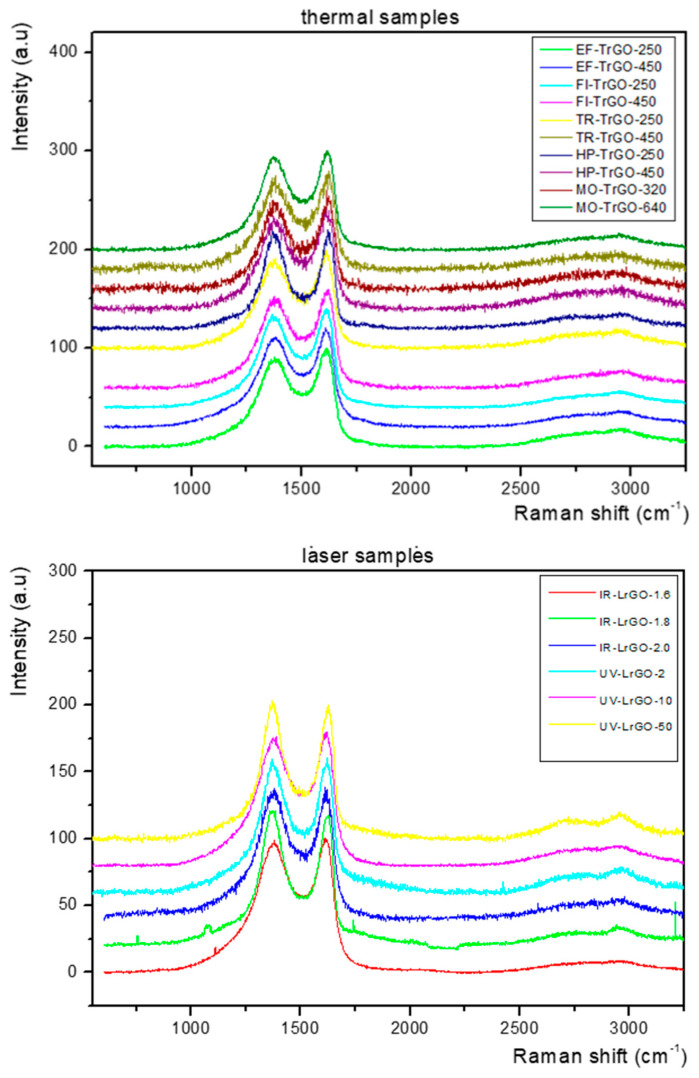
Raman spectra corresponding to the TrGO (**top**) and LrGO (**bottom**) samples. All spectra were normalized to the intensity of the G band.

**Figure 6 nanomaterials-13-01391-f006:**
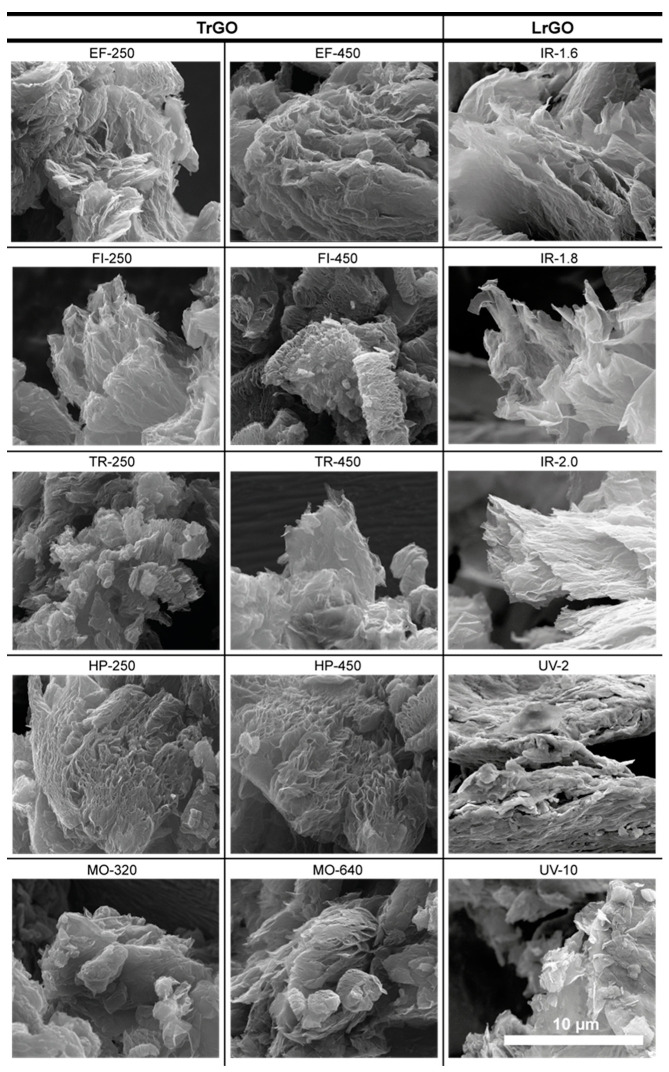
SEM images of TrGO and LrGO representative samples. Scale bar shown at bottom right applies to all images.

**Table 1 nanomaterials-13-01391-t001:** Nomenclature of the rGO samples.

Method (y)	Instrument (x)	Parameter (z)		
		Temperature (°C)	Power (W)	Exposure Time (ms)
Thermal reduction x-TrGO-z	Electrical furnace EF-y-z	EF-TrGO-250		
		EF-TrGO-450		
	Fusion instrument FI-y-z	FI-TrGO-250		
		FI-TrGO-450		
	Tubular reactor TR-y-z	TR-TrGO-250		
		TR-TrGO-450		
	Heating plate HP-y-z	HP-TrGO-250		
		HP-TrGO-450		
	Microwave oven MO-y-z		MO-TrGO-320	
			MO-TrGO-640	
Laser reduction x-LrGO-z	IR laser (10.6 µm) IR-y-z		IR-LrGO-1.6	
			IR-LrGO-1.8	
			IR-LrGO-2.0	
	UV laser (405 nm) UV-y-z			UV-LrGO-2
				UV-LrGO-10
				UV-LrGO-50

**Table 2 nanomaterials-13-01391-t002:** Comparison of structural parameters resulting from the XRD analysis of the samples. FWHM: full width at half-maximum, in degrees (deg); d: average distance between rGO layers; D1: average height of each sample stacking nanolayer for peak (002), with constant 0.9; n: average number of rGO layers in the graphene stacking nanolayer; D2: average diameter of each sample stacking nanolayer for peak (100), with constant 1.84.

Sample	(002) Peak				(100) Peak				
	2θ	FWHM 2θ	d (nm)	D1 (nm)	n	2θ	FWHM 2θ	d (nm)	D2 (nm)
EF-TrGO-250	23.3	4.54	0.39	4	10	43.3	2.5	0.210	14
EF-TrGO-450	24.3	3.40	0.37	5	13	43.0	2.8	0.210	13
FI-TrGO-250	23.5	4.13	0.38	4	11	43.3	2.4	0.209	15
FI-TrGO-450	23.7	3.67	0.37	5	12	43.0	2.6	0.211	13
TR-TrGO-250	23.7	4.94	0.40	3	9	43.3	2.7	0.211	13
TR-TrGO-450	24.6	5.54	0.38	3	8	43.2	2.6	0.211	14
HP-TrGO-250	23.3	4.82	0.39	4	9	43.1	2.4	0.211	15
HP-TrGO-450	23.9	5.78	0.37	3	8	43.6	2.7	0.207	13
MO-TrGO-320	23.5	3.84	0.38	4	12	42.3	2.0	0.214	18
MO-TrGO-640	24.5	4.62	0.36	4	10	43.3	2.5	0.211	14
IR-LrGO-1.6	25.9	1.07	0.34	16	46	42.2	0.80	0.214	43
IR-LrGO-1.8	25.8	1.74	0.34	10	28	42.3	0.80	0.214	43
IR-LrGO-2.0	25.8	0.94	0.34	18	53	42.2	0.94	0.214	37
UV-LrGO-2	26.6	0.30	0.33	57	169	42.4	0.49	0.213	71
UV-LrGO-10	26.6	0.27	0.34	64	190	42.4	0.49	0.213	71
UV-LrGO-50	26.4	0.54	0.34	32	94	43.3	0.49	0.209	71

**Table 3 nanomaterials-13-01391-t003:** Textural characterization of TrGO and LrGO samples.

Sample	SSA (m^2^ g^−1^)	Micropore Volume (cm^3^ g^−1^)
EF-TrGO-250	391	0.020
EF-TrGO-450	468	0.044
FI-TrGO-250	450	0.024
FI-TrGO-450	553	0.032
TR-TrGO-250	340	0.016
TR-TrGO-450	373	0.023
HP-TrGO-250	244	0.0070
HP-TrGO-450	682	0.025
MO-TrGO-320	39	0.0040
MO-TrGO-640	322	0.021
IR-LrGO-1.6	68	0.0026
IR-LrGO-1.8	84	0.0019
IR-LrGO-2.0	75	0.0013
UV-LrGO-2	49	0.0080
UV-LrGO-10	107	0.010
UV-LrGO-50	133	0.034

**Table 4 nanomaterials-13-01391-t004:** I_D_/I_G_ and I_2D_/I_G_ ratio in Raman analysis.

Sample	I_D_/I_G_	I_2D_/I_G_
EF-TrGO-250	1.008	0.189
EF-TrGO-450	1.044	0.168
FI-TrGO-250	1.020	0.158
FI-TrGO-450	0.979	0.169
TR-TrGO-250	1.008	0.178
TR-TrGO-450	1.023	0.152
HP-TrGO-250	1.056	0.139
HP-TrGO-450	0.964	0.198
MO-TrGO-320	0.975	0.169
MO-TrGO-640	1.035	0.152
IR-LrGO-1.6	1.143	0.092
IR-LrGO-1.8	1.028	0.093
IR-LrGO-2.0	1.233	0.169
UV-LrGO-2	1.040	0.182
UV-LrGO-10	1.067	0.152

## Data Availability

Data are contained within the article.

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
