# Peer review of "Comparison of Thermal and Laser-Reduced Graphene Oxide Production for Energy Storage Applications"

_nanomaterials, 2023, doi:10.3390/nano13081391_

Round 1
Reviewer 1 Report
Comments to the Author:
Belén Gómez-Mancebo et al. study the thermal and laser reduced graphene oxide and inn this study, five different heating methods were used for the thermal reduction, and UV and CO2 lasers were used for the photothermal and/or photochemical reduction. The chemical and structural characterizations of the fabricated rGO samples were performed by BET, XRD, SEM and Raman spectroscopy measurements. The analysis and comparison of the results revealed that the strongest feature of the thermal reduction methods is the production of high specific surface area, fundamental for volumetric energy applications such as hydrogen storage; whereas in the case of the laser reduction methods, a highly localized reduction is achieved, ideal for microsupercapacitors in flexible electronics.
Though this work has obtained promising results, many aspects still need to be improved. Therefore, the following issues are recommended for further justification and clarification.
1. From the figure 2, the reduced GO powder is clearly observed. Hence, are the SEM results in figure 6 also based on powder? What is the size of samples prepared for SEM?
2. Five different heating process as well as equipment are chosen for thermal reduction, and XRD result from each sample in figure 3 shows that the MO-TrGO 320 has the prominent peaks. Could authors explain the origin of these different results? For example, why microwave oven can give the sample with obvious peaks, and why there is better result when the power is 320W rather than 640W?
3. In figure 5 (bottom), the Raman result shows miscellaneous peaks which are explained as the presence of remnant functional groups or defects, however, authors have not stated the reason why the laser reduced GO has these defects.
4. Authors have given a detailed characterization of reduced GO and expect the application of it. However, authors have not exhibited any contrast with rGO from other research and any possible application examples. Hence, reviewer recommends authors added some reference to support their statement.
5. Besides the research works containing parameters of nonlinear refraction coefficient. several papers will also help authors to analyze the applications of 2D materials. The papers with a recent study on 2D materials can be parts of the reference of this manuscript, which will help to fill in the details (Wan, Z et al Materials & Design 2021, 212, 110185; Haoran Mu et al 2022 Mater. Futures 1 012301; Rui Xu et al 2022 Mater. Futures 1 032302, Yan, Z et al Small 2022, 18, 2200016).
Author Response
Referee 1 Belén Gómez-Mancebo et al. study the thermal and laser reduced graphene oxide and inn this study, five different heating methods were used for the thermal reduction, and UV and CO2 lasers were used for the photothermal and/or photochemical reduction. The chemical and structural characterizations of the fabricated rGO samples were performed by BET, XRD, SEM and Raman spectroscopy measurements. The analysis and comparison of the results revealed that the strongest feature of the thermal reduction methods is the production of high specific surface area, fundamental for volumetric energy applications such as hydrogen storage; whereas in the case of the laser reduction methods, a highly localized reduction is achieved, ideal for microsupercapacitors in flexible electronics.
Though this work has obtained promising results, many aspects still need to be improved. Therefore, the following issues are recommended for further justification and clarification.
We thank the reviewer for the valuable comments and suggestions. We have answered them point-by-point. All changes are indicated within the answers.
- From the figure 2, the reduced GO powder is clearly observed. Hence, are the SEM results in figure 6 also based on powder? What is the size of samples prepared for SEM?
All SEM images were taken using the same SEM system specified in the methods section of the article. Regarding the sample preparation, all graphene oxide samples, transformed using several laser and thermal techniques were collected as powder. A small amount of each powder (1 mm2) was placed on a large area SEM sample holder using tweezers, and with the help of standard double-sided conductive carbon tape to avoid charge accumulation on the sample during imaging.
2. Five different heating process as well as equipment are chosen for thermal reduction, and XRD result from each sample in figure 3 shows that the MO-TrGO 320 has the prominent peaks. Could authors explain the origin of these different results? For example, why microwave oven can give the sample with obvious peaks, and why there is better result when the power is 320W rather than 640W?
The prominent peak seen from the sample MO-TrGO-320 corresponds to the one centered at ≅10°, which corresponds to the (001) peak of 12° of GO. This means that it is the opposite: the reduction degree, and therefore the amount of carbon, is higher in the case of 640W than 320W because the graphene oxide peak has almost disappeared under 640W. This is confirmed further from table 2: the value of 2θ for the sample MO-TrGO-640 corresponding to the (002) peak is of 24.5°, whereas 2θ value for MO-TrGO-320 is of 23.5°. Thus, the value of 2θ for the sample MO-TrGO-640 is closer to the one corresponding to graphite: 26°. In the same manner, the interlayer spacing of 0.36 nm from MO-TrGO-640 is closer to the one from graphite (0.34 nm) than the one of 0.38 nm, corresponding to the sample MO-TrGO-320. All these results prove that a better reduced GO is obtained under 640W.
A new paragraph has been added in the text on page 6 accordingly: “In samples MO-TrGO-320, UV-LrGO-2 and UV-LrGO-10, the presence of a more prominent peak at ≅10° (2ÆŸ) corresponds to (001) GO reflection. This is due to a lower degree of reduction under these conditions”.
3. In figure 5 (bottom), the Raman result shows miscellaneous peaks which are explained as the presence of remnant functional groups or defects, however, authors have not stated the reason why the laser reduced GO has these defects.
We appreciate the comment from the reviewer: the sentence was too general and incomplete. Therefore, we have added a more detailed analysis to the original text: “The degree of resolution of the 2D and D+G peaks varies slightly depending on the sample; however, the high level of broadness and convolution of the peaks indicates the multilayered structuring of the rGO materials produced and the presence of remnant functional groups or defects [37, 38].”
Now this text has been rewritten accordingly in page 13:
“The resolution level of the 2D and D+G peaks varies slightly depending on the sample; however, the high level of broadness and convolution of the peaks indicates the multilayered structuring of the produced rGO materials and the presence of remnant functional groups and defects [54, 55]. The remnant -OH, -H and -O functional groups are related to the degree of reduction, which is modulated by the processing conditions: temperature, time, atmosphere, wavelength and power. The incomplete removal of these functional groups leads to the existence of lattice disorder and consequently, Raman peaks associated to defects [56]. On the other hand, the sudden release of the CO and CO2 gases during the heating helps the exfoliation into thin graphene-like sheets. However, it also generates structural defects due to the high amount of pressure exerted over the material in a short period of time [57]. Finally, in the case of the laser reduced graphene oxide materials, the preparation method of the samples, namely the scratching from the substrate after reduction to obtain the powder, could have also contributed to the creation of structural defects. The reduction in the intensity of the Raman peaks related to defects, the increase in the intensity of the 2D peak and the narrowing of the FWHM of the D, G, 2D and D+G peaks can be achieved optimizing further the reduction parameters of the seven methods employed in the present work.”
The new references are written below and have been added as:
[56] Krishnamoorty, K., Veerapandian, M., Yun, K., and Kim S. J. “The chemical and structural analysis of graphene oxide with different degrees of oxidation”. Carbon 2013, 53, 38-49, https://doi.org/10.1016/j.carbon.2012.10.013
[57] Pei, S., and Cheng, H. M. “The reduction of graphene oxide.” Carbon 2012 50. 9, 3210-3228, https://doi.org/10.1016/j.carbon.2011.11.010
4. Authors have given a detailed characterization of reduced GO and expect the application of it. However, authors have not exhibited any contrast with rGO from other research and any possible application examples. Hence, reviewer recommends authors added some reference to support their statement. We have strengthened the introduction by adding two new paragraphs, citing relevant literature from the field: regarding the reduction mechanisms and both applications, hydrogen storage and flexible microsupercapacitors.
Paragraph 1:
“According to the literature, thermal and laser annealing of GO can effectively remove functional groups depending on the applied temperature and the other experimental conditions such as time, power and pressure. Between 200 and 550 ËšC hydroxyl and partially epoxy and carboxyl groups are removed [16]. Increasing the annealing temperature results in higher C/O ratios than those obtained using other reduction strategies [17]. This highly reduced graphene oxide allow to restore the conjugated structure and usually exhibits improved charge transport capabilities [18]. This makes the highly reduced graphene oxide suitable for high-quality electronic devices [19], sensors [20] and energy storage [21]. In addition, most reduction strategies require that GO must be dispersed in a solvent where sheets can freely move and they can be close to each other. This usually led to agglomeration and layer stacking if spacers such as nano-fillers are not used [22]. In contrast, in thermal and laser reduction GO sheets are dispersed in solid phase, which reduces the extent of the sheets agglomeration [23]. In summary, thermal and laser reductions are easier and comparatively less expensive methods to obtain rGO that make them a feasible route for the bulk production of graphene.”
Paragraph 2:
“Previous works have validated the suitability of the thermal and laser reduction methods for the fabrication of graphene-based materials for energy storage applications, specifically hydrogen storage [24-26] and flexible microsupercapacitors [27-30]. In the present work, the correlation between the fabrication methods and the properties of the produced materials enables to determine which method is more suitable for each of the afore mentioned applications: the bulk synthesis nature of the thermal methods leads to the production of materials with higher SSA, ideal for volumetric applications such as hydrogen storage. The point-by-point, highly localized and superficial nature of the laser methods is ideal for in-plane and miniaturized applications such as flexible microsupercapacitors.”
5. Besides the research works containing parameters of nonlinear refraction coefficient. several papers will also help authors to analyze the applications of 2D materials. The papers with a recent study on 2D materials can be parts of the reference of this manuscript, which will help to fill in the details (Wan, Z et al Materials & Design 2021, 212, 110185; Haoran Mu et al 2022 Mater. Futures 1 012301; Rui Xu et al 2022 Mater. Futures 1 032302, Yan, Z et al Small 2022, 18, 2200016). We want to thank the reviewer for the valuable tips and the suggestion of these references. We have included them together with an additional analysis in the text, in pages 8 and 9:
“These results suggest that oxygen and water existing between the layers could be removed through both thermal and laser processes, which, in addition to reducing GO to rGO, facilitates exfoliation of bulk rGO into few-layer rGO, as happened in other materials to be used as 2D materials [Haoran Mu et al 2022 Mater. Futures 1 012301] (ref. 34)”
“Taking into account all the parameters calculated by XRD, TrGO materials consist of a few graphene layers, in the range of 8 to 13, denoted n in Table 2, in a stacking nanostructure of an average diameter by height of approximately 4 nm x 14 nm, and a graphene layer distance of 0.38 nm. The numbers of layers obtained in TrGO samples issimilar to those obtained in other 2D materials [Yan, Z et al Small 2022, 18, 2200016] (ref 37)”
“There is a significant difference between the two lasers as shown in Table 2 with smaller stacking nanostructure in IR-LrGO samples. These IR-LrGO samples show similar numbers of layers to other vertical stacking heterostructures for optoelectronic devices [Wan, Z et al Materials & Design 2021, 212, 110185]. (ref 38)”

Reviewer 2 Report
The paper “Comparison of Thermal and Laser Reduced Graphene Oxide 2 Production for Energy Storage Applications” studies the effects of thermal and laser reduction on graphene oxide using a range of thermal treatments and two different laser setups. The produced materials are characterized by a range of techniques like XRD, BET, Raman and SEM.
The paper is well written and conclusions are well supported by data. However the introduction does not account for existing literature on thermal and laser reduction of graphene oxide with makes the novelty of the paper not clear. I thus recommend the authors revise the introduction accordingly, citing the relevant literature on thermal and laser reduction of graphene oxide and highlighting more clearly the novelty of this paper
Other minor points to address are listed below:
The authors report scan speeds of the reducing laser up to 600 mm/s which looks incredibly fast for a mechanical stage to achieve. Please check these numbers or justify how it can be reached
In line 173 the authors mention a resolution of 327 ppi. Please define ppi
In the same paragraph the authors describe a process to make samples for laser reduction (UV) which is different from the process describe for the previous laser reduction (IR). So I wonder why different processes are needed in these two cases and please clarify
In lines 308-313 the authors observe differences in the material properties between thermal shock and temperature ramp. Please comment on that as to why you think this happens
Figure 5 shows raman spectra which are better shown normalized to appreciate differences between spectra
Author Response
Referee 2 We thank the reviewer for the valuable comments and suggestions. We have answered them point-by-point. All changes are indicated within the answers.
The paper is well written and conclusions are well supported by data. However the introduction does not account for existing literature on thermal and laser reduction of graphene oxide with makes the novelty of the paper not clear. I thus recommend the authors revise the introduction accordingly, citing the relevant literature on thermal and laser reduction of graphene oxide and highlighting more clearly the novelty of this paper
We have strengthened the introduction by adding two new paragraphs where the novelty and motivation of this work are better highlighted. These two paragraphs include new citations of relevant literature from the field.
Paragraph 1:
“According to the literature, thermal and laser annealing of GO can effectively remove functional groups depending on the applied temperature and the other experimental conditions such as time, power and pressure. Between 200 and 550 ËšC hydroxyl and partially epoxy and carboxyl groups are removed [16]. Increasing the annealing temperature results in higher C/O ratios than those obtained using other reduction strategies [17]. This highly reduced graphene oxide allows to restore the conjugated structure and usually exhibits improved charge transport capabilities [18]. This makes the highly reduced graphene oxide suitable for high-quality electronic devices [19], sensors [20] and energy storage [21]. In addition, most reduction strategies require that GO must be dispersed in a solvent where sheets can freely move and they can be close to each other. This usually led to agglomeration and layer stacking if spacers such as nano-fillers are not used [22]. In contrast, in thermal and laser reduction GO sheets are dispersed in solid phase, which reduces the extent of the sheets agglomeration [23]. In summary, thermal and laser reductions are easier and comparatively less expensive methods to obtain rGO that make them a feasible route for the bulk production of graphene.”
Paragraph 2:
“Previous works have validated the suitability of the thermal and laser reduction methods for the fabrication of graphene-based materials for energy storage applications, specifically hydrogen storage [24-26] and flexible microsupercapacitors [27-30]. In the present work, the correlation between the fabrication methods and the properties of the produced materials enables to determine which method is more suitable for each of the afore mentioned applications: the bulk synthesis nature of the thermal methods leads to the production of materials with higher SSA, ideal for volumetric applications such as hydrogen storage. The point-by-point, highly localized and superficial nature of the laser methods is ideal for in-plane and miniaturized applications such as flexible microsupercapacitors.”
Other minor points to address are listed below:
The authors report scan speeds of the reducing laser up to 600 mm/s which looks incredibly fast for a mechanical stage to achieve. Please check these numbers or justify how it can be reached
The machine used is a heavy-duty machine which can reach the cited moving speeds as specified by the manufacturer. The sample is fixed, mounted on a static plate and typically fixed with tape. Only the laser head moves at the specified speed, however, the used values rarely go over 100 mm/s.
In line 173 the authors mention a resolution of 327 ppi. Please define ppi
In this case, this metric is related to the control of the UV laser, and refers to “pixels per inch”, the resolution of a printing or engraving process. This UV laser machine, designed to engrave images on arbitrary surfaces using a UV laser, is controlled in the same parameters. In this case, 327 pixels per inch refer to a pixel size of 75 μm.
We have added this information in the page 5, paragraph 5.
In the same paragraph the authors describe a process to make samples for laser reduction (UV) which is different from the process describe for the previous laser reduction (IR). So I wonder why different processes are needed in these two cases and please clarify
Yes, as the machines that control each laser are different, also the sample size that can be introduced in the machines are different. In the case of the UV laser, the reduced dimensions of the machine require for smaller samples. However, the amount of deposited material was calculated to produce the same thickness in proportion with the deposited area.
In lines 308-313 the authors observe differences in the material properties between thermal shock and temperature ramp. Please comment on that as to why you think this happens
A paragraph has been introduced in the text on page 10, explaining this point.
“When a thermal shock occurs in these materials or when a temperature ramp is applied in which a higher temperature is reached (in this case there is also a greater temperature difference) the emission of CO and CO2 is very violent, which favours the expansion of the material and as a consequence increases the SSA. In cases where 250°C is applied, with a temperature ramp, the applied temperature difference is lower, therefore the emission of CO and CO2 is less steep and the rGO exfoliation is less effective.”
Figure 5 shows raman spectra which are better shown normalized to appreciate differences between spectra
All spectra were already normalized to the intensity of the G band. This sentence has been introduced for clarification in the text, in the caption to figure 5, page. 13:
“Figure 5. Raman spectra corresponding to the TrGO (top) and LrGO (bottom) samples. All spectra were normalized to the intensity of the G band.”

Round 2
Reviewer 2 Report
I acknowledge that the authors have significantly improved the paper and addressed my points satisfactorily.
However, some of the points that were addressed were not translated into the text, so I recommend the following replies are reflected into the text:
“The sample is fixed, mounted on a static plate and typically fixed with tape. Only the laser head moves at the specified speed, however, the used values rarely go over 100 mm/s.”
“as the machines that control each laser are different, also the sample size that can be introduced in the machines are different. In the case of the UV laser, the reduced dimensions of the machine require for smaller samples. However, the amount of deposited material was calculated to produce the same thickness in proportion with the deposited area.”
Finally, as for the Raman spectra shown in Fig. 5b, the second spectrum from the bottom (IR-LrGO-1.8) seems not normalized as it overlaps with the bottom one (IR-LrGO-1.6). Please fix this.
After these minor aspects are fixed I recommend publication on Nanomaterials
Author Response
I acknowledge that the authors have significantly improved the paper and addressed my points satisfactorily.
However, some of the points that were addressed were not translated into the text, so I recommend the following replies are reflected into the text:
“The sample is fixed, mounted on a static plate and typically fixed with tape. Only the laser head moves at the specified speed, however, the used values rarely go over 100 mm/s.”
“as the machines that control each laser are different, also the sample size that can be introduced in the machines are different. In the case of the UV laser, the reduced dimensions of the machine require for smaller samples. However, the amount of deposited material was calculated to produce the same thickness in proportion with the deposited area.”
We would like to thank the referee for these suggestions.
We have added the two paragraphs where they correspond in the text in page 5
Finally, as for the Raman spectra shown in Fig. 5b, the second spectrum from the bottom (IR-LrGO-1.8) seems not normalized as it overlaps with the bottom one (IR-LrGO-1.6). Please fix this.
We have changed the Fig. 5b, for the corrected one with all the normalized spectra.
After these minor aspects are fixed I recommend publication on Nanomaterials
